# Modelling the impacts of marine heatwaves on plankton in the Salish Sea

Karyn D. Suchy<sup>1</sup>, Susan E. Allen<sup>1</sup>, Akash R. Sastri<sup>2,3</sup>, and Kelly Young<sup>2</sup>

Department of Earth, Ocean and Atmospheric Sciences, University of British Columbia, Vancouver, BC, V6T 1Z4, Canada Institute of Ocean Sciences, Fisheries and Oceans Canada, Sidney, BC, V8L 5T5, Canada

Correspondence to: Karyn D. Suchy (ksuchy@eoas.ubc.ca)

Abstract. Marine heatwaves are discrete events of prolonged anomalously warm ocean temperatures caused by a combination of atmospheric forcing and ocean processes. The Northeast Pacific Marine Heatwave (NEP-MHW) was first detected in the Salish Sea in 2014 and persisted in the region until 2017. Here, we used a three-dimensional coupled biophysical model, SalishSeaCast, to examine the impacts of the NEP-MHW on the physics and the plankton in the Salish Sea. Sixteen years (2007-2022) of model results were used to follow the trajectory of the NEP-MHW into key regions of the Salish Sea. Model results were compared to observation data collected over the same period. We resolved the specific impacts of the NEP-MHW versus the impacts of warming via other large-scale climate indices operating on longer time scales. Model results showed that the strongest physical signatures of the NEP-MHW were evident in the Juan de Fuca region wherein warming was favourable for the growth of both diatoms and nanoflagellates. In comparison, the direct warming from the NEP-MHW impacted the Strait of Georgia (SoG) to a lesser degree but warm water anomalies persisted in this region until the end of our study period in 2022. Both temperature and nitrate in the upper layer of the SoG were strongly linked to the North Pacific Gyre Oscillation and diatom biomass decreased during this prolonged warming period. Our results highlight the need to recognize that multiple types of marine heatwaves associated with different large-scale climate indices can occur simultaneously, even within a single waterbody such as the Salish Sea, each with distinct impacts on the local food web.

# 1 Introduction

Marine heatwaves (MHWs) are discrete events of prolonged (i.e., longer than 5 days) and anomalously warm ocean temperatures in a given region (Pearce et al., 2011; Hobday et al., 2016) often caused by a combination of atmospheric forcing and/or ocean processes (Oliver et al., 2021). Large-scale climate patterns have been shown to influence the likelihood that a MHW event will occur (Scannell et al., 2016; Holbrook et al., 2019). For example, the El Niño Southern Oscillation (ENSO) is the dominant mode of interannual climate variability worldwide and the leading cause of MHW occurrences (Oliver et al., 2018). Globally, MHW events have increased in frequency and duration (Oliver et al., 2018) and are expected to further increase as a result of long-term ocean warming (Frölicher et al., 2018).

<sup>&</sup>lt;sup>3</sup>Department of Biology, University of Victoria, Victoria, BC, V8W 3N5, Canada

The Northeast Pacific Marine Heatwave (NEP-MHW), initially called "The Blob" (Bond et al., 2015), developed in the Gulf of Alaska during the boreal winter of 2013 and was the largest MHW on record in the region (Di Lorenzo and Mantua, 2016). The development of the NEP-MHW was attributed to strong positive anomalies in sea level pressure, which reduced winds and ultimately suppressed the heat flux from the ocean to the atmosphere (Bond et al., 2015). Later, the cause of this MHW was attributed to ocean heat transport, as opposed to anomalous air-sea heat fluxes into the ocean (Oliver et al., 2021), and thus linked to tropical-extratropical teleconnections (Bond et al., 2015; Di Lorenzo and Mantua, 2016; Oliver et al., 2021). More recently, Chen et al., (2023) showed that warm air temperatures played the most significant role in the development of the NEP-MHW. Although there is no clear consensus on how the NEP-MHW was formed, the North Pacific Gyre Oscillation (NPGO) and the Pacific Decadal Oscillation (PDO) were shown to reflect temperature and sea level pressure changes during this time period (Di Lorenzo and Mantua, 2016). Overall, the likelihood of a MHW occurring in the NE Pacific is elevated by about 17% during years that are in a positive PDO phase, yet ENSO and NPGO are also known to play significant roles in the occurrences of MHW in this region (Holbrook et al., 2019).

The NEP-MHW had widespread ecosystem impacts in the North Pacific. For example, lower chlorophyll *a* concentration (a proxy for phytoplankton biomass) and a shift to a higher relative abundance of picoplankton as a result of temperature-induced changes to biological processes were observed near Ocean Station Papa (Yang et al., 2018). Similarly, high SST anomalies, fresher nearshore water, and anomalous periods of downwelling were associated with a decrease in chlorophyll *a* in the Southern California Current System (Zaba and Rudnick, 2016). With respect to zooplankton, the copepod community off of the coast of Oregon shifted to smaller, lipid-poor taxa, which decreased the quality of food available to higher trophic levels (Peterson et al., 2017). Similarly, the abundance of the smaller copepod *Calanus pacificus*, which is typically most abundant off the coasts of California and southern Oregon, increased throughout the Northeast Pacific (Fisher et al., 2020). The NEP-MHW also significantly impacted many commercially important shellfish and finfish fisheries because of a widespread coastal toxic bloom of the diatom *Pseudo-nitzschia* (McCabe et al., 2016). A mass mortality event of common murres (*Uria aalge*) as well as other fish, bird, and mammal species was observed throughout the region from 2014-2017 (Piatt et al., 2020).

The Salish Sea is a large coastal waterbody in the southwest portion of British Columbia, Canada, and northwest portion of Washington, USA. The strongest reported impacts of the NEP-MHW within the Salish Sea appeared in autumn 2014 and persisted through the 2015-2016 El Niño wherein anomalous water column warming was observed (Chandler et al., 2017; Khangaonkar et al., 2021). Khangaonkar et al., (2021) used a modelled simulation of the NEP-MHW period and found a small decrease in both phytoplankton and zooplankton biomass relative to reference conditions in the Salish Sea. Observed zooplankton biomass was anomalously high in all regions within the U.S. waters of the Salish Sea during 2015, remaining high until 2017 in Admiralty Inlet and Central Basin (Winans et al., 2023). Biomass of some of the dominant zooplankton taxa (e.g., copepods, euphausiids, and gelatinous groups) showed variable and basin-specific responses during NEP-MHW years,

70

but these patterns may have been related more to local drivers rather than due to warming from NEP-MHW (Winans et al., 2023). In contrast, zooplankton biomass was below average in 2014 in the Strait of Georgia (Canadian waters; Perry et al., 2021). Further, the increased water temperatures during the NEP-MHW in Haro Strait were associated with decreased body condition of age-0 Pacific sand lance (*Ammodytes personatus*), an important forage species for piscivorous seabirds (Robinson et al., 2023).

Here, we used a three-dimensional coupled biophysical model, SalishSeaCast, to examine the impacts of the NEP-MHW on the physics and the plankton in the Salish Sea. Sixteen years (2007-2022) of model results were used to follow the trajectory of the NEP-MHW into key regions of the Salish Sea to determine the extent to which this event impacted lower trophic level dynamics. Given that other warming phenomena (e.g., negative phase of the NPGO and El Niño events) were co-occurring over the study period, we aimed to resolve the specific impacts of the NEP-MHW versus the impacts associated with other large-scale climate indices often operating on longer time scales in the region. Temperature and nitrate were analyzed over the full water column as well as averaged over the relevant depth layers associated with estuarine circulation in the Salish Sea to track incoming water. Phytoplankton and zooplankton were then examined over the upper water column. Model results were then ground-truthed with observation data collected over the same period. Results from this study were interpreted and discussed in the context of the multiple warming phenomena operating during the same time in the Salish Sea. Since MHWs are projected to increase in frequency, there is a need to understand their complex impacts on marine ecosystems.

# 2 Methods

90

# 2.1 Study Site

The Salish Sea is a waterbody comprised of four major areas: the Strait of Georgia (SoG), Puget Sound (PS), Haro Strait, and Juan de Fuca Strait (JdF) which connects the region to open ocean waters (Fig. 1). The main source of freshwater into the region is the Fraser River, resulting in an estuarine-like circulation with surface waters leaving the SoG via the JDF (Pawlowicz et al., 2007). The roughly 100 km long JdF simultaneously brings in estuarine inflow waters at depth (100-200 m) carrying offshore waters which are high in nutrients and low in oxygen (Pawlowicz et al., 2007). Average flushing time, a measurement of water exchange, in the SoG is 125 days compared to 47 days in PS (MacCready et al., 2021). Water entering the SoG at depth during estuarine exchange is comprised of a combination of the intermediate water being brought into the JdF and surface SoG water that has been mixed extensively in the Haro Strait region (Ianson et al., 2016). During the summer months, 70% of the water entering the Salish Sea via the JdF comes from north and offshore Pacific sources (Beutel and Allen, 2024). However, in winter a combination of southern shelf water and, to a lesser extent, Columbia River plume water, makes up over 80% of the inflow (Beutel and Allen, 2024). Our study focused on five main regions of interest: JdF, Haro Strait, Central Basin in Puget Sound, Central SoG, and Northern SoG (Fig. 1) in order to follow the trajectory of the NEP-MHW of 2014-2017 and its potential impacts on plankton in the Salish Sea.

Figure 1: Map of the Salish Sea study area. Model domain is indicated with the light grey box. Selected sub-regions for analysis are shown in blue boxes Bathymetry is overlain to highlight regional differences.

## 2.2 Study Period

Our modelled study period was from 2007 to 2022 (Fig. 2). The NEP-MHW formed during winter 2013 into 2014 (Bond et al., 2015) and persisted through to 2017, yet several other warming signals were present during this same period. In this paper we consider two additional large-scale climate signals: the North Pacific Gyre Oscillation (NPGO), which fluctuates on approximately decadal timescales, and the Southern Oscillation Index (SOI) operating on shorter (2-7 year) time scales. Negative values of both NPGO and SOI (El Niño conditions) have previously been associated with positive SST anomalies, high Fraser River discharge, and weak winds in the SoG (Suchy et al., 2019). Further, strong relationships have been shown between the NPGO and SOI and plankton dynamics within the Salish Sea (Suchy et al., 2022; Suchy et al., 2025a). The NPGO represents the second empirical orthogonal function (EOF) of monthly residual SST anomalies in the North Pacific (Di Lorenzo et al., 2008). The SOI is calculated as pressure differences between the tropical eastern and western Pacific, providing a traditional measure of El Niño and La Niña events in the Pacific Ocean (Wallace et al., 1998). Monthly NPGO Index data were downloaded from (www.o3d.org/npgo/npgo.php) and monthly SOI Index data were downloaded from

(www.bom.gov.au/climate/current/soi2.shtml). In addition, a resurgence of positive sea surface temperature (SST) anomalies
 referred to as the Blob 2.0 appeared in the summer of 2019 in the Northeast Pacific (Amaya et al., 2020). Our results for the NEP-MHW of 2014 to 2017 are considered within the context of these other climate signals which were also operating over the modelled period.

Figure 2: (a) Monthly North Pacific Gyre Oscillation (NPGO) index and Southern Oscillation Index (SOI) values during the modelled study period of 2007 to 2022, (b) monthly wind speed anomalies calculated over the Central Strait of Georgia. Wind data were unavailable for December 2015, and (c) monthly anomalies in Fraser River Discharge. The Northeast Pacific marine heatwave of 2014-2017 is indicated with pink shading.

## 125 2.3 SalishSeaCast Model

The SalishSeaCast model domain covers the entire Salish Sea (Fig. 1) with a horizontal resolution of approximately 500 m and a vertical resolution ranging from 1 at the surface to 27 m at 400 m depth. The physical component of SalishSeaCast is an

implementation of Nucleus for European Modelling of the Ocean (NEMO Version 3.6; Madec et al., 2017) and is described in detail in Soontiens et al., (2016) and Soontiens and Allen, (2017) with subsequent relevant changes outlined in Olson et al., (2020); Jarníková et al., (2022); and Stang and Allen, (2025). Comprehensive evaluations of SalishSeaCast model have already been completed for nitrate (Olson et al., 2020), chlorophyll (Olson et al., 2020; Jarníková et al., 2022), and zooplankton (Suchy et al., 2023). This study was based on v202111 of SalishSeaCast. The initial conditions for v202111 are from the previous model version v201905 (Suchy et al., 2023; 2025). The model was spun up for five years prior to the first model year of 2007 to allow variables to reach equilibrium.

135

140


The model uses real-time data from Environment and Climate Change Canada (ECCC) as input for the Fraser River discharge (Soontiens and Allen, 2017) in addition to daily river flows for over 150 other rivers in the region (Stang and Allen, 2025). Atmospheric forcing (winds, solar radiation) is derived from High Resolution Deterministic Prediction System (HRDPS) atmospheric model output (Milbrandt et al., 2016). SalishSeaCast has two open boundaries for temperature, salinity, and nutrients: a northern boundary at Johnstone Strait and a western boundary at the mouth of Juan de Fuca Strait. Prior to 2013, western boundary conditions were based on fields from NEP 3.6 (Lu et al., 2017). After 2013, open boundary conditions were based on fields from the LiveOcean model (Davis et al., 2014; Siedlecki et al., 2015). Northern boundary conditions are based on temperature and salinity climatologies (Dosser et al., 2021).



The biological component of the model, SMELT (Salish Sea Model Ecosystem-Lower Trophic), follows the transfer of the model's currency (nitrogen) between nutrients, primary producers, grazers, and detrital pools with coupled silicon and oxygen cycling (Olson et al., 2020). The nutrients in the model are nitrate, ammonium, and silicon. Differences between v202111 and the previous model version (v201905; Jarníková et al., 2022) are provided in the supplemental information (Tables S1-S9). The ciliate group (*M. rubrum*) was removed due to its consistently low contribution to overall phytoplankton biomass and the lack of improvement in model performance when previously included. Therefore, two groups of primary producers are present in the current version of the model: diatoms and nanoflagellates. Functional light dependence was switched to a potential energy curve but tuned to match the old response closely. The sinking calculation for biological tracers switched from upstream to incorporation in the Total Variation Diminishing (TVD) advection. Lastly, the nitrogen to oxygen coupling for various processes was updated and a parameter for sediment oxygen demand, proportional to the amount of organic matter sinking out of the model domain, was added to effectively allow an oxygen flux into the sediments uncoupled to outgoing nitrate flux. Dissolved silica was set to a concentration of 120 μm Si in Puget Sound based on data provided by the Washington Department of Ecology (2021) and was kept at 59.57 μm Si in all other rivers (Olson et al., 2020).



Diatoms in the model have the highest maximum growth rates, the highest optimal light requirements, and are the only class to take up silicon (Olson et al., 2020). Nanoflagellates have the lowest maximum growth rate but compete better at low nitrogen concentrations and high temperatures (Olson et al., 2020; Jarníková et al., 2022). A previous evaluation of the model





phytoplankton classes against high performance liquid chromatography (HPLC) data from the Canadian waters of the Salish Sea (Nemcek et al., 2023) showed that larger, centric diatoms are well represented by the model diatom class, whereas the model nanoflagellate class showed the strongest relationships with cryptophytes, prasinophytes, and haptophytes (Suchy et al., 2025a).

The temperature response for each phytoplankton group is set so that the optimal temperature for growth for diatoms (12°C) and nanoflagellates (18°C) match those of diatoms and dinoflagellates in Khangaonkar et al., (2012), after model experiments with these settings showed improved summer chlorophyll bias. Diatoms become nitrate-limited at a half-saturation constant of 2.0 µM N, whereas the half-saturation constant prescribed for nanoflagellates is 0.1 µm N. Additionally, the model includes biogenic silica, detrital particulate organic nitrogen (PON), and dissolved organic nitrogen (DON). Phytoplankton growth in the model may be limited by temperature, light, or nutrients (nitrate/ammonium, silicon).

Heterotrophs in SalishSeaCast are represented by two zooplankton classes: zooplankton class 1 (Z1) and zooplankton class 2 (Z2). Identification of which zooplankton taxa are represented by Z1 or Z2 was determined by evaluations against observation data (Suchy et al., 2023). The Z1 class freely evolves based on model dynamics (Olson et al., 2020) and represents taxa with traits such as relatively small body size, short life cycles, or high grazing rates, and whose growth rates respond quickly to local conditions (e.g., non-overwintering copepods, copepod nauplii, larvaceans; Suchy et al., 2023). Z2 are the highest trophic level whose grazing impact is included in the model and represents taxa with larger body size, longer life cycles, and slower grazing rates (e.g., overwintering copepods, euphausiids, amphipods; Suchy et al., 2023). The domain-mean Z2 biomass is constrained due to a seasonal climatology imposed as part of the model closure (see Suchy et al., 2023) and the Z2 biomass is distributed spatially throughout the model domain in proportion to food availability. Thus, spatial variability in the distribution of Z2 throughout the domain will directly reflect differences in the spatial distribution of Z2 prey items (diatoms, nanoflagellates, Z1, and PON). Olson et al., (2020) provides a full description of the biological model for version 201812 of SalishSeaCast. Subsequent changes, including those affecting zooplankton rates, were made for v201905 (Jarníková et al., 2022; Suchy et al., 2023; 2025). Adjustments made to the biology for the current version of the model (v202111) are provided in Supplemental Tables S2-S4.

#### 2.4 Model Data

We analyzed 16 years (2007-2022) of monthly model output from SalishSeaCast. Model data were averaged across each of the five study regions (blue boxes in Fig. 1). All model data are presented as monthly anomalies, which were calculated by subtracting the climatological mean (i.e., the mean for each month averaged over all 16 years) from the value for each month within a given year. Model output for Conservative Temperature (Θ; hereafter referred to as temperature in °C) and nitrate are shown with time versus depth plots, as well as being averaged over specific depth ranges (0-50m, 50-150m, 150m to bottom;




mmol N m<sup>-3</sup>). We focus our discussion mainly on the surface (0-50m) as this depth layer is most relevant to both phytoplankton and zooplankton. Information for deeper layers is provided in the Supplemental Information. Halocline strength, a proxy for water column stratification, was calculated as the difference in salinity divided by the difference in depth of the two model grid cells wherein the maximum salinity gradient was observed and provides an indication of how much energy is required to mix the water column. Hourly wind data from HRDPS were interpolated onto the model grid and then calculated as mean monthly wind speed values for the Central SoG region only. Daily Fraser River discharge data from 2007-2022 were obtained from Environment and Climate Change Canada (www.wsc.ec.gc.ca/applications/H20/index-eng.cfm) from Station 08MF005 at Hope, BC, and then used to calculate monthly anomalies. Model output for phytoplankton (diatoms and nanoflagellates) and zooplankton (Z1 and Z2) biomass were depth-averaged over the 0-50 m depth range to capture the full extent of the euphotic zone across regions (mmol N m<sup>-3</sup>). In addition, the extent to which temperature dependence and light/nutrient limitation was limiting to growth was calculated based on the phototrophic growth rate equations in the model (Suchy et al., 2025a) and shown with anomaly plots. Pearson Product-Moment Correlations were used to analyze the relationships between temperature and nitrate anomalies versus the NPGO and SOI.

#### 2.5 Observation Data

Nitrate, chlorophyll a (Chl a), and zooplankton data were obtained from numerous sources to compare our model results with observations in the JdF and Central SoG regions. Nitrate and Chl a data were initially compiled by Parker MacCready (University of Washington; https://github.com/parkermac/LO/tree/main/obs). Zooplankton observation data were provided by the Institute of Ocean Sciences, Fisheries and Oceans Canada, and are described in detail for the model evaluations presented in Suchy et al., (2023). The zooplankton data used in this study can be downloaded from (https://data.cioospacific.ca). Model and observation data were averaged over the same approximate regions. To maximize the number of model-observation comparisons, nitrate and Chl a were averaged over the 0-10 and 0-50 m depth ranges, respectively, whereas zooplankton were averaged over the full water column for both model and observations. Mean annual values for nitrate, Chl a, and zooplankton were then calculated over pre-MHW (2007-2013), MHW (2014-2019 to capture the NEP-MHW and the Blob 2.0), and post-MHW (2020-2022) years for comparison with model data across the same time periods. Model units (mmol N m<sup>-3</sup>) were converted to biomass units using a Chl:N ratio of 2 for phytoplankton (mg m<sup>-3</sup>) and a C:N ratio of 4.9 for zooplankton (mg C m<sup>-3</sup>; see Suchy et al., 2023).

## 3 Results & Discussion

#### 3.1 Physical Drivers

### **225 3.1.1 Temperature**



Overall, the strongest positive temperature anomalies occurred during two time periods across all regions: during the 2009-2010 El Niño event and during the NEP-MHW years; however, regional variability in the strength of these anomalies was evident (Figs. 3, 4). Positive temperature anomalies of up to  $0.83^{\circ}$ C were observed in the 0-50 m (surface) depth layer in JdF from the end of 2009 through to the end of 2010 (El Niño event; Figs. 3a, 4a). The 2010 El Niño event was also evident in the SoG with positive temperature anomalies occurring mid-2009 and throughout 2010 in the Central and Northern SoG. Unlike the JdF region, positive anomalies in the 0-50m layer in the SoG were periodically interrupted with intrusions of cooler water at the end of 2009 (Fig. 4). A period of negative temperature anomalies then followed the 2010 El Niño event in all regions, persisting until the end of 2013 (Fig. 4).

Strong positive temperature anomalies appeared in the surface waters of the JdF region in mid-2014, reaching 1.8°C above the mean by the end of the year and persisting through 2017, consistent with the NEP-MHW years (Figs. 3a,4a). After mid-2019, negative temperature anomalies in the JdF region returned until the end of our time series in 2022. Temperature anomalies in the 50-100 m (intermediate) and 150 m to bottom (deep) depth layers were similar to those observed for the upper water column except that the warm anomalies in these layers arrived in pulses during the winter months from 2014-2017, and then again in the winter of 2019 (Figs. S1, S2) when a second MHW occurred in the Northeast Pacific (Amaya et al., 2020; Chen et al., 2021). Temperature anomalies in Haro Strait and Puget Sound were similar to those observed in JdF at all depth layers; however, the surface (0-50 m) layer reached maxima of 1.4°C and 1.3°C above the mean in Haro Strait and Puget Sound, respectively, suggesting this region exhibited slightly less warming compared to JdF (Figs. 3,4, S1-S2). Our results are consistent with another modelling study from the Salish Sea (Khangaonkar et al., 2021), where depth-averaged temperature increases of up to 1.5°C were observed in Puget Sound during the NEP-MHW years despite the use of different reference years (i.e., 2013 versus the climatological mean used in the present study).

Our model results showed that positive temperature anomalies in the SoG appeared at the end of 2013 and thus earlier than the NEP-MHW reported for coastal regions, but the timing was consistent with the initiation of the NEP-MHW in the Gulf of Alaska (Bond et al., 2015). Therefore, it is likely that the increased warming in this region originated from other sources such as atmospheric heating or local wind anomalies. Positive temperature anomalies in the 0-50 m depth layer reached maximum values of 1.0°C and 0.9°C in early 2015 in the Central and Northern SoG, respectively, which were the lowest warming anomalies out of any of our study regions during NEP-MHW years (Fig. 4). Most notably, the Central SoG remained warm

Figure 3: Model-based temperature anomalies versus depth in the different sub-regions of the Salish Sea from 2007 to 2022. Red indicates positive temperature anomalies and blue represents negative temperature anomalies. The Northeast Pacific marine heatwave of 2014-2017 is outlined with dashed grey lines. Note that the vertical (depth) scales vary, reflecting the maximum depth of each region.


Figure 4: Mean model-based 0-50 m temperature anomalies in the different sub-regions of the Salish Sea from 2007 to 2022. Temperature anomalies for the 50-150m and 150m to bottom depth ranges are provided in supplemental material (Figs. S1-S2).

until the end of 2022, indicating a longer-term warming signal than can be explained by the NEP-MHW alone. Significant relationships were found between surface temperature anomalies in the Central SoG and the NPGO (r = -0.36, p 

2022; Suchy et al., 2025a). In contrast, surface temperature anomalies in the JdF region were significantly correlated to the SOI (r = -0.44, p < 0.001), but not to the NPGO (r = -0.11, p = 0.12; Table 1).

Table 1: Pearson Product-Moment Correlations used to assess the relationship between temperature and nitrate anomalies versus large-scale climate indices (North Pacific Gyre Oscillation; NPGO and Southern Oscillation Index; SOI) for the Juan de Fuca and Central Strait of Georgia regions. Statistically significant relationships are indicated in bold with the following annotations: \* = p < 0.05, \*\* = p < 0.01, \*\*\* = p < 0.001.

| Region Name  |               | Temperature Anomalies |          |          |
|--------------|---------------|-----------------------|----------|----------|
|              | Climate Index | 0-50 m                | 50-150   | >150 m   |
| Juan de Fuca | NPGO          | -0.11                 | -0.01    | 0.14     |
|              | SOI           | -0.44***              | -0.43*** | -0.38*** |
| Central SoG  | NPGO          | -0.36***              | -0.30*** | -0.08    |
|              | SOI           | -0.33***              | -0.38*** | -0.28*** |
|              |               | Nitrate Anomalies     |          |          |
|              |               | 0-50 m                | 50-150 m | >150 m   |
| Juan de Fuca | NPGO          | 0.08                  | 0.12     | 0.15*    |
|              | SOI           | 0.35***               | 0.40***  | 0.36***  |
| Central SoG  | NPGO          | 0.48***               | 0.57***  | 0.64***  |
|              | SOI           | 0.11                  | 0.23**   | 0.33**   |



We note here that the intermediate (50-150 m) and deep (>150 m) layers in the Central and Northern SoG showed extended warming during both the 2010 El Niño and the 2015 El Niño/NEP-MHW (Figs. S1-S2). Positive temperature anomalies were evident in these depth layers until early 2011, followed by a cooler period until the end of 2014. After the arrival of the NEP-MHW in 2014, these layers stayed warm until 2020 when negative temperature anomalies returned until the end of 2022. The prolonged warming observed in the deeper waters of the SoG was shown in a previous modelling study of the region (Khangaonkar et al., 2021) as well as in an observation study from Rivers Inlet, BC (Jackson et al., 2018). Jackson et al., (2018) suggested that warm water anomalies are only removed from deep fjords via either mixing with colder waters above or when colder water enters over the sill. In the present study, the model results suggest that mixing of cooler water from above did not occur since the 0-50 m depth layer remained warm following the NEP-MHW in the Central SoG. However, cooling of



the deep basin waters of the Central SoG was evident once negative temperature anomalies were present in the JdF intermediate layers (Fig. S1), which eventually flow into the SoG (Pawlowicz et al., 2007; Allen et al., 2025).

## 3.1.2 Halocline Strength

The Central SoG and PS regions had the strongest haloclines (i.e., more stratification) whereas haloclines in the JdF and Haro Strait regions were weak (more mixed; Fig. 5). From 2007-2014 halocline strength periodically showed positive anomalies during spring and summer months in the Central SoG, which corresponded to periods of low winds in the Central SoG region and high Fraser River discharge (Fig. 2b,c; Suchy et al., 2025a). Positive anomalies in halocline strength were evident at the end of 2014 and into early 2015 in these regions and then periodically throughout the NEP-MHW years (Fig. 5). Negative anomalies in halocline strength during this time corresponded to negative anomalies in Fraser River discharge and positive wind anomalies in the Central SoG (Fig. 2b,c). Although negative halocline anomalies occurred throughout 2019 in Puget Sound, strong stratification signals persisted in the Central SoG until the end of 2022 most likely a result of the weak winds that were prevalent from 2013 until the end of our study period (Fig. 2b).

## 3.1.3 Nitrate

Nitrate anomalies were mostly consistent across all depth layers (0-50 m, 50-150 m, and >150 m; Figs. 6-7, S3-S4). Positive nitrate anomalies were observed in all regions from 2007 until approximately mid-2013 with the exception of a slight decrease in nitrate during the winter/early spring of the 2010 El Niño in JdF, Haro Strait, and Puget Sound (Fig. 6). The strongest negative nitrate anomalies occurred in the JdF region during the NEP-MHW years (Figs. 6-7). In comparison, the Central SoG had the weakest negative nitrate anomalies during the NEP-MHW years wherein nitrate was periodically replenished into the surface waters (Fig. 6d,7d), likely due to wind-induced upwelling events (Moore-Maley and Allen, 2022). Periodic increases in nitrate were also observed at depth in the Central SoG in 2017, which may have been a result of either advection from Haro Strait or regeneration of nitrate in the deeper waters within the SoG (Sutton et al., 2013). Following the NEP-MHW years of 2014-2017, two distinct patterns were observed: nitrate anomalies returned to predominantly positive values in early 2020 in the JdF, Haro Strait, and PS regions; in contrast, nitrate anomalies in the Central and Northern SoG remained predominantly negative until end of 2022. Surface nitrate anomalies were significantly correlated with the NPGO in the Central SoG (*r* = 0.48, *p* 

Figure 5: Anomalies in mean model-based halocline strength (a proxy for stratification) in the different sub-regions of the Salish Sea from 2007 to 2022.

Figure 6: Model-based nitrate anomalies versus depth in the different sub-regions of the Salish Sea from 2007 to 2022. Red indicates positive temperature anomalies and blue represents negative temperature anomalies. The Northeast Pacific marine heatwave of 2014-2017 is outlined with dashed grey lines. Note that the vertical (depth) scales vary, reflecting the maximum depth of each region. White line is missing data from June 2008.

Figure 7: Mean model-based 0-50 m nitrate anomalies in the different sub-regions of the Salish Sea from 2007 to 2022. Nitrate anomalies for the 50-150m and 150m to bottom depth ranges are provided in supplemental material (Figs. S3-S4).

# 3.2 Different Types of Heatwaves


Overall, our results suggest that the main physical signatures of the NEP-MHW reached as far as Haro Strait and PS from the JdF, but the warming response in the surface waters of the SoG was muted and likely forced by independent factors affecting







local conditions. We show here that different types of warming events existed simultaneously within a relatively small geographic region. The regional analysis presented in this study revealed that warming may be driven by atmospheric changes and large-scale climate patterns acting either simultaneously (e.g., El Niño events corresponding with the NEP-MHW years in this study) or operating somewhat independently of the NEP-MHW (e.g., the NPGO exerting the strongest influence on local conditions).

The strongest temperature and nitrate anomalies in all regions were observed during the NEP-MHW years. Positive temperature anomalies at all depth layers in the JdF, Haro Strait, and PS regions, as well as deeper layers (below 50 m) in the Central and Northern SoG, also corresponded with El Niño events. Furthermore, the nitrate signal in all regions except for the SoG were typically lower during El Niño years. In general, upwelling-favourable winds (equatorward) are present along the west coast of Vancouver Island from late March to September (Thomson, 1981). During winter (October through early March) there is an abrupt reversal of the prevailing winds producing downwelling-favourable conditions; however, these typical conditions can be significantly altered by El Niño events (Harris et al., 2009). Globally, marine heatwaves associated with El Niño events have been shown to suppress nutrients in some regions by weakening the upwelling of nutrient-rich waters (Hayashida et al., 2020). Indeed, the association of El Niño events with anomalously weak upwelling and a subsequent reduction in nutrients has been observed in both the California Current System (Jacox et al., 2015) and off the west coast of Vancouver Island (Harris et al., 2009). Since upwelling fluxes are the largest offshore source of nutrients near the mouth of Juan de Fuca Strait (Peña et al., 2019), we expected the effects of variability in upwelling to propagate into the Salish Sea via the source of water entering through the intermediate depth layer (Beutel and Allen, 2024). Focusing specifically on winter, water temperatures during the winter months in the Northeast Pacific were found to be warmer and less nutrient-rich during an El Niño event (Whitney et al., 1998), which supports our findings of winter intrusions of warm, lower-nitrate waters in the JdF, Haro Strait, and PS regions during the stronger El Niño events (negative SOI) of 2009-2010 and 2015-2016 (Fig. 2). Therefore, the temperature and nitrate patterns propagating into the JdF, Haro Strait, and Puget Sound regions are likely a result of a combined El Niño and NEP-MHW effect.

Conversely, the SoG heatwave was associated with the NPGO signal more strongly than with the NEP-MHW, and possibly even suppressing effects of the NEP-MHW. Mean monthly temperature anomalies for the surface (0-50 m) depth layer, as well as nitrate concentrations at all depth layers, in the Central and Northern SoG were tightly coupled with the NPGO (Table 1). Monthly values for the NPGO index indicated a shift to the negative (warm) phase in October 2013 (Fig. 2), which corresponded to the shift to negative nitrate anomalies in the SoG (Figs. 6-7). The NPGO is associated with fluctuations in salinity, nutrients, and chlorophyll *a* concentrations in the Northeast Pacific (Di Lorenzo et al., 2008). Previously, negative nitrate anomalies in the Central SoG were linked to weaker winds during negative NPGO years as shown in Figs 2b and 7, which prevented nutrients from being mixed into the surface waters during warm years (Suchy et al., 2025a). The contrasting pattern we found for the SoG deeper water nitrate signal compared to other regions after the NEP-MHW can be explained by

the unique ways in which nitrate is entrained back into and/or recycled in the deep basin of the strait. Specifically, due to the intense mixing in the Haro Strait region, a large portion of the water leaving the SoG eventually gets mixed back into the region (Pawlowicz et al., 2007; Ianson et al., 2016; Allen et al., 2025). In addition, due to the high biological productivity in the SoG, nitrate is constantly being regenerated in the region (Ianson et al., 2016; Sutton et al., 2013).

## 390 3.3 Impacts of Marine Heatwaves on Plankton

To discuss the impacts of warming on phytoplankton and zooplankton in the Salish Sea we focused our results on two regions representative of the different types of heatwaves observed: i) the JdF region, which was strongly influenced by the co-occurrence of the 2014-2017 NEP-MHW and El Niño-related warming events, and ii) the Central SoG wherein nitrate was strongly coupled to the NPGO, thus masking the weaker effects of the 2014-2017 NEP-MHW. These two regions are separated by strong tidal mixing in Haro Strait and are influenced by distinct physical drivers of plankton dynamics (Suchy et al., 2023) discussed below.

## 3.3.1 Phytoplankton




During the NEP-MHW, prolonged positive anomalies in diatom concentration were seen at the beginning of 2015 in the JdF region; however stronger positive anomalies in diatoms persisted from 2018 to 2020 (Fig. 8a, Supp. Fig. S5a). In addition, a strong positive anomaly (44% increase compared to the mean) in nanoflagellates, the concentration of which is typically relatively low in JdF, was observed in 2015 (Fig. 8b, Supp. Fig. S5b). Diatoms and nanoflagellates in the JdF region experience cooler temperatures compared to the other regions (climatological mean and maximum water column temperatures of 7.1 and 12.3°C in this study, respectively). Thus, the warming of up to 2°C during the NEP-MHW in this region was favourable for the growth of both groups, bringing conditions closer to their optimal temperatures (set in the model at 12°C and 18°C for diatoms and nanoflagellates, respectively) compared to the climatology.

Anomalies of phytoplankton can be positive or negative in response to marine heatwaves depending on which resource (e.g., light, nutrients) is the most strongly limiting within a given study region (Hayashida et al., 2020). Indeed, diatom growth in the JdF region was less limited by temperature due to the warming during the 2010 El Niño and the NEP-MHW years of 2014-2017 (positive values for the purple lines; Fig. 9a). Light limitation on diatom growth was more variable with slightly less light limitation occurring from 2014-2017 compared to other years in the study. In addition, while the JdF region is generally nutrient replete year round, the decrease in nitrate associated with the NEP-MHW years resulted in a slight increase in nitrate and silicon limitation on diatom growth compared to the climatology (negative values for blue and green lines; Fig. 9c), but this limitation was not strong enough to inhibit photosynthetic growth of diatoms at any point during the study period in the JdF region.

Figure 8: Anomalies in mean model-based 0-50 m biological parameters (diatoms, nanoflagellates, Z1, and Z2 zooplankton in the Juan de Fuca and Central Strait of Georgia regions from 2007 to 2022. For anomalies including the seasonal cycle see Fig. S5.

In contrast, diatom concentrations decreased and nanoflagellates increased in the Central SoG during the NEP-MHW years with negative diatom anomalies continuing to persist for longer durations each year until the end of 2022 (Fig. 8b,d; Supp. Fig. S6a,b). Also evident was that diatoms during the post-2014 period (i.e., during negative or warm-phase NPGO years) peaked earlier, and for shorter durations, before switching to nanoflagellate dominance compared to the 2007 to 2014 period (Fig. 8b, Supp. Fig. S6a). In general, the Central SoG is the region within the Salish Sea with the warmest water temperatures


(mean and maximum climatological full water column temperatures of 9.9 and 19.5°C, respectively). Our limitation plots indicated that diatom growth in this region was occasionally limited by temperature (Fig. 9c); however, the Central SoG periodically experienced negative temperature anomalies in the 0-50 m depth layer during the NEP-MHW of 2014-2017 (Fig. 4d), thus temporarily alleviating any temperature limitation during the main NEP-MHW years. Overall, nitrate was the most limiting to diatom growth during the summer months throughout the study, with persistent nitrate limitation occurring in the upper 50 m during all seasons from 2017 to 2022 (blue line negative values; Fig. 9d), consistent with previous findings that this region has the strongest nitrate limitation out of any region in the Salish Sea due to its strong stratification and high biological productivity (Suchy et al., 2023).

Figure 9. Anomalies in the temperature dependence, light and nitrate limitation on diatom growth in the 0-50 m depth layer compared to the 16-year climatology (2007-2022) in the Juan de Fuca (a,b) and Central Strait of Georgia (b,c) regions. Positive values mean less limitation or temperature dependence compared to climatology; negative values reflect more limitation or temperature dependence.






## 3.3.2 Zooplankton

Anomalies in Z1 and Z2 zooplankton concentrations followed patterns like those observed for diatom anomalies in the JdF region (Fig. 8, Supp. Figs. S5). No clear variations in either of the zooplankton classes were observed in relation to the NEP-MHW years; however, stronger positive anomalies of Z1 zooplankton occurred from about 2014 to 2020. This pattern was also evident for Z2 zooplankton in the JdF where mostly negative anomalies persisted from 2007 to 2014 and then switched to mostly positive anomalies for the remainder of the study period (Fig. 8g). Z1 anomalies closely followed the diatom pattern in the Central SoG, but negative anomalies in Z2 concentration predominated from 2014 to 2022 (Figs. 8f,h, Supp. Fig. 6), coinciding with negative NPGO years. Given that spatial variability in Z2 biomass in the model reflects differences in the spatial distribution of their prey, our results suggest that zooplankton experienced better feeding conditions in the JdF region compared to the Central SoG during the latter half of our study period.

## 3.4 Model and Observation Comparisons

A well-tuned model that has been evaluated against observation data can provide an unprecedented level of information to fisheries managers, biologists, and other relevant parties. Here, we assessed the model's ability to represent real world conditions by comparing mean monthly SalishSeaCast model results to monthly observation data averaged across the JdF and Central SoG regions (Figs. 10 & 11). In JdF, our results showed that the model did not capture the range of observed 0-10 m nitrate concentrations; however, the seasonal cycles of model versus observations were comparable (Fig. 10a). The range of nitrate concentrations in the model was 12-26 μM whereas observation values ranged from 5-30 μM, including very low observation values during the NEP-MHW years. Seasonal peaks in Chl a in the observations were generally captured by the model (Fig. 10b). Observed Chl a ranged between 0.2 and 12 mg m<sup>-3</sup> but model chlorophyll had a smaller range of 0.6 to 3 mg m<sup>-3</sup>. Unfortunately, few observation zooplankton samples were available in the JdF region prior to 2014, which limited our comparison (Fig. 10c). That said, model zooplankton (Z1 and Z2 combined) values were always within the range of observed values.

Both model nitrate and chlorophyll *a* showed better agreement with the observed values in the Central SoG even though the model failed to capture the very high chlorophyll *a* peaks that occurred throughout the observation time series (Fig. 11a,b). Observation zooplankton data were more comprehensive in this region, thus allowing for more accurate ground truthing. Our results showed that model zooplankton biomass was higher than observations over the course of our study period (Fig. 11c). These results were expected as the model represents zooplankton that we know are not captured adequately in traditional zooplankton nets (see Suchy et al., 2023). Better agreement between model and observations was evident after 2015 corresponding to increased temporal resolution of observation sampling (Fig. 11c).

Figure 10: SalishSeaCast Model comparison with observation data for the Juan de Fuca region for a) 0-10 m nitrate, b) 0-50 m chlorophyll a, and c) full water column zooplankton. Chlorophyll observation data were compared to the sum of model diatoms and nanoflagellates multiplied by a Chl:N of 2. Zooplankton observation data were compared to the sum of model Z1 and Z2 zooplankton biomass. Dark blue lines indicate mean model values with combined spatial (grid point) and temporal (daily) standard deviations in light blue shading. Symbols for observation data are the mean monthly value with whiskers showing standard deviations.

Figure 11: SalishSeaCast Model comparison with observation data for the Central SoG region for a) 0-10 m nitrate, b) 0-50 m chlorophyll a, and c) full water column zooplankton. Chlorophyll observation data were compared to the sum of model diatoms and nanoflagellates multiplied by a Chl:N of 2. Zooplankton observation data were compared to the sum of model Z1 and Z2 zooplankton biomass. Dark blue lines indicate mean model values with combined spatial (grid point) and temporal (daily) standard deviations in light blue shading. Symbols for observation data are the mean monthly value with whiskers showing standard deviations.

Our attempts to determine if the marine heatwave-related patterns we found for the model were also evident in the observation data were met with some challenges. For example, to compare with observation data we needed to compromise on both spatial

and temporal scales due to differences in sampling resolution. Spatially, we expanded the extent of the JdF and Central SoG observations beyond the initial model boxes provided in Fig. 1 to ensure an adequate number of observations were available for comparison with model results (Fig. S7). Temporally, since observation data were not consistently available at the monthly spatial resolution used for model output, particularly in the JdF region, we had to use annual mean values for comparisons between the observation and model datasets prior to grouping into pre-MHW, MHW, and post-MHW years.

Significant differences in nitrate in the JdF region were found between pre-MHW, MHW, and post-MHW years for model nitrate ( $\mathbf{F}(2, 13) = 4.59, p 

Figure 12: Mean 0-10 m nitrate concentration, 0-50 m chlorophyll a biomass, and full water column zooplankton biomass premarine heatwave years (Pre MHW; 2007-2013), marine heatwave years (MHW; 2014-2019), and post-marine heatwave years (Post MHW; 2020-2022) for observation and data in the Juan de Fuca Strait region. Asterisk indicates significant differences at α = 0.05.

Vertical scales differ between observed and modelled chlorophyll a and zooplankton to account for the higher and more variable observations, as shown in previous model evaluations (Olson et al. 2020, Suchy et al. 2023).

Figure 13: Mean 0-10 m nitrate concentration, 0-50 m chlorophyll a biomass, and full water column zooplankton biomass premarine heatwave years (Pre MHW; 2007-2013), marine heatwave years (MHW; 2014-2019), and post-marine heatwave years (Post MHW; 2020-2022) for observation and data in Central SoG region. Asterisk indicates significant differences at  $\alpha = 0.05$ . Vertical scales differ between observed and modelled chlorophyll a and zooplankton to account for the higher and more variable observations, as shown in previous model evaluations (Olson et al. 2020, Suchy et al. 2023).

## **4 Conclusions**

555

Our model results showed that the strongest physical signatures of the NEP-MHW were evident in the JdF region, followed by Haro Strait and Puget Sound. The positive temperature anomalies and negative nitrate anomalies characteristic of the NEP-MHW years were also present in these regions during the El Niño of 2009-2010. Given that the waters in the JdF region are,

560

565

on average, cooler and more nutrient-replete compared to other regions, the increased temperatures associated with the NEP-MHW favoured phytoplankton growth and a resulted in a subsequent increase in zooplankton biomass, while the decrease in nitrate had little to no effect on the phytoplankton. The JdF region experienced intrusions of cooler water at depths >50 m both during and after the NEP-MWH. In contrast, warming associated with the NEP-MHW had less of an impact on plankton in the Central SoG. Although positive temperature anomalies were prevalent during the NEP-MHW years in this region, they were weaker compared to other regions and remained anomalous in the surface until the end of our study period. Colder water only entered the deep layers of the SoG after consistent negative temperature anomalies were observed in the intermediate waters of the JdF region. In contrast to the JdF region, both temperature and nitrate were strongly linked to the NPGO index in the Central SoG. A decrease in diatom concentration was observed during NPGO negative (warm) years, including the NEP-MHW years, likely due to the nitrate limitation associated with weaker wind-driven resupply of nutrients to the surface waters. Comparison of model and observation data highlighted a common challenge faced by researchers attempting to use a combined model-observation approach: mismatches in spatial and temporal scales led to limitation in our analyses. Nevertheless, the findings presented here also highlight the need for comprehensive in situ sampling programs as model tuning improves with increased observation data availability. The regional analysis presented in this study revealed that multiple types of marine heatwaves can impact different sub-regions within the same coastal water body and, as a result, have unique repercussions throughout the food web.

# 5 Code and Data Availability

575 SalishSeaCast model results (version 202111) and model forcing fields are available online: (http://salishsea.eos.ubc.ca/erddap/griddap/index.html). The model code for NEMO-3.6 is available from the NEMO website (www.nemo-ocean.eu; Madec et al., 2017). The Jupyter Notebooks used for model output and analysis in this paper are available on GitHub preserved at https://doi.org/10.5281/zenodo.17584029 (Suchy et al., 2025b).

# 580 6 Author Contributions

KDS performed the analyses and drafted the initial manuscript. SEA performed the hindcast simulations of SalishSeaCast. ARS and KY performed observation data curation. KDS and SEA acquired financial support for the project leading to this publication. All authors contributed equally to the development of the research concept and to the manuscript beyond the initial draft.

# 7 Competing Interests

The authors declare that they have no conflict of interest.

# 8 Acknowledgements

This work was funded by the British Columbia Salmon Restoration and Innovation Fund of Fisheries and Oceans, Canada (grant #BCSRIF\_2022\_358). Computational resources for SalishSeaCast are provided by Digital Research Alliance of Canada, Ocean Networks Canada, and Advanced Research Computing and the Department of Earth, Ocean and Atmospheric Sciences both of the University of British Columbia. The SalishSeaCast model software environment was developed by Doug Latornell. We thank Becca Beutel for assistance with compiling observation data for temperature, nitrate, and chlorophyll *a*.

595

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
