# Peer review of "Modelling the impacts of marine heatwaves on plankton in the Salish Sea"

_EGUsphere, 2025_

## Referee Comment (RC2)

**Summary**

This manuscript describes the impacts of marine heatwaves on different regions of the Salish Sea. It examines the relationships between the Northeast Pacific marine heatwave (NEP-MHV; 2014–2017) and larger-scale climate signals such as the SOI and the NPGO. The study investigates the effects of temperature anomalies during these heatwave events on regional ecosystems, highlighting cascading effects across different phytoplankton and zooplankton groups. The results show that both the drivers of these perturbations and their ecological impacts vary among regions. The study combines modeling and observations, a comprehensive and relevant approach that supports the validity of the SalishSeaCast model, which performs well when compared with observations.

Overall, this work contributes meaningfully to our understanding of ecosystem perturbations associated with extreme climate events, an issue of growing importance as marine heatwaves are becoming more frequent and intense under ongoing global warming.

My main concern with the manuscript relates to the interpretation of the results. The processes underlying the contrasted responses observed across regions are complex but crucial to understand and to highlight. In its current form, these processes are not sufficiently interpreted or discussed, and the manuscript would benefit from a more thorough discussion of the mechanisms behind the observed patterns. The manuscript is well structured, clearly written, and highly relevant to the scope of the journal, provided that the following comments are addressed.

**General comments**

The manuscript does not include a dedicated Discussion section; elements of discussion are integrated in the Results section. However, interpretation of the impacts of the different marine heatwaves on biogeochemistry and lower trophic levels should be addressed in a dedicated section. It is essential to discuss the processes linking large-scale climate signals and local MHWs, as well as the relationships between temperature and nitrate anomalies and phytoplankton and zooplankton responses, including differences among the functional groups or the model.

The Discussion should address the following specific points:

- o Why are temperature anomalies in different regions more strongly correlated with certain large-scale climate signals than others (SOI, NPGO). The manuscript would benefit from presenting hypotheses, supported by relevant literature, to interpret these correlations.

- For both sites (JdF and SoG), the manuscript should explain the cascading processes by which temperature and nitrate anomalies propagate through the food web, from phytoplankton to zooplankton, and discuss why/how these pathways differ between sites.

- Interpretation and discussion of the zooplankton results presented in Section 3.3.2 (see related comment below).

**Specific comments**

**Section 2.2 Study Period**

- Throughout the manuscript, you refer to extended time periods associated with well-known large-scale climate signals, such as the NEP-MHV (2014–2017), SOI and El Niño events (e.g., 2010, 2015), and NPGO years (post-2014). However, these periods are described using different expressions, which may be confusing for the reader (e.g., L246: "*post-2014 period (i.e., during negative or warm-phase NPGO years)*"; L382, L453: "*negative NPGO years*"; L565: "*NPGO negative (warm) years*").
  To improve clarity and consistency, it would be helpful to explicitly describe these climate events when they are first introduced in section 2.2, and to define a clear terminology that is then used throughout the manuscript. This approach is already well implemented for the NEP-MHV period (L103), and a similar treatment for the other climate signals would greatly enhance readability. For example: "*Monthly values for the NPGO index indicated a shift to the negative (warm) phase in October 2013 (Fig. 2). Hereafter, we refer to the post-2014 period as the 'negative NPGO years'.*"

- In several sections, key statements are supported almost exclusively by self-citations and are not sufficiently placed within the broader literature. For instance, in Section 2.2 (L108–108), relationships between NPGO/SOI and physical and biological conditions in the Salish Sea are referenced primarily through the authors' previous work, and in Section 3.1 (L274), relationships between temperature and large-scale climate indices in the Strait of Georgia are supported in a similar way. While these references are clearly relevant, including additional independent studies would strengthen the manuscript by better situating the results within the broader scientific context.

**Section 2.3 SalishSeaCast Model**

- L152: *"Functional light dependence was switched to a potential energy curve but tuned to match the old response closely"*.
  What do you mean by *"tuned"*? Please elaborate.

- L174-187: Could you briefly describe the trophic interactions among the different model components (diatoms, nanophytoplankton, Z1, Z2)? This information would be useful for interpreting the cascading effects observed during marine heatwaves.

**Section 2.4 Model Data**

- L195: *"We focus our discussion mainly on the surface (0-50m) as this depth layer is most relevant to both phytoplankton and zooplankton."*, and L202: "*Model output for phytoplankton (diatoms and nanoflagellates) and zooplankton (Z1 and Z2) biomass were depth-averaged over the 0-50 m depth range to capture the full extent of the euphotic zone across regions (mmol N m-3)"*.
  The manuscript would benefit from including information on the vertical structure and migratory behavior of zooplankton in the different regions of the Salish Sea. First, this would support the assumption that a 0–50 m layer is appropriate for representing zooplankton. In addition, such information would be valuable for interpreting the results on the impacts of marine heatwaves on lower trophic levels (cf. my comment on Section 3.3.2).

- L204: "*the extent to which temperature dependence and light/nutrient limitation was limiting to growth was calculated based on the phototrophic growth rate equations in the model*".
  Could you please elaborate on the methodology of this diagnostic?

**Section 3.1.1 Temperature**

- L271: "*indicating a longer-term warming signal than **can** be explained by the NEP-MHW alone*". Can or can't ?

- L275: "*In contrast, surface temperature anomalies in the JdF region were significantly correlated to the SOI (r = -0.44, p < 0.001), but not to the NPGO (r = -0.11, p = 0.12; Table 1).*" Could you provide an interpretation of this result, or propose hypotheses to explain it? See the general comment regarding the need for a dedicated Discussion section.

**Section 3.3.1. Phytoplankton**

- L411: "*slightly less light limitation*".
  Please consider providing a metric or quantitative indicator to better support the interpretation of Figure 9. Moreover, why was this analysis/diagnostic performed only for diatoms? Applying a similar diagnostic to all four model components (diatoms, Z1, Z2) would help to interpret the processes underlying the variation patterns observed across the different trophic levels studied.

- L426: "*Also evident was that diatoms during the post-2014 period (i.e., during negative or warm-phase NPGO years) peaked earlier, and for shorter durations, before switching to nanoflagellate dominance compared to the 2007 to 2014 period (Fig. 8b, Supp. Fig. S6a).* »
  Unclear, please clarify or rephrase. In particular, what is meant by *"peaked earlier"*, earlier relative to what?

**Section 3.3.2. Zooplankton**

- In the JdF region, the marine heatwave has been described as being linked to the NEP-MHV and SOI, with the end of positive temperature anomalies and negative nitrate anomalies around 2020. From 2020 to 2022, negative temperature anomalies reappear, along with positive nitrate anomalies. A similar shift is observed for both nanophytoplankton and diatoms, with predominantly negative anomalies over the 2020–2022 period. Consequently, negative anomalies are also observed for Z1, but we do not observe the same response of Z2 (except for a peak in 2022). How can these contrasting responses between Z1 and Z2 be explained?
  More generally, providing information and relevant literature on the trophic structure and/or taxonomic composition in the different regions studied would greatly help to interpret these results. In particular, differences between the SoG and JdF sites, as well as communities' changes across the different periods within each site, should be discussed. Overall, this part of the results would benefit from being accompanied by interpretation in the Discussion section.

**Section 3.3.4. Model and Observation Comparisons**

- L467: "*model zooplankton (Z1 and Z2 combined) values were always within the range of observed values*".
  Figure 10c shows that the model does not capture the maximum observed zooplankton values. Please clarify.

- L470: *"Both model nitrate and chlorophyll a showed better agreement with the observed values in the Central SoG"*.
  Computing a comparison metric, such as the RMSE, for all variables in the observation–model comparison would help support this statement. This is not obvious for chlorophyll-a

- L486: Please reconsider the scale of Figure 11c, using a range of approximately 0–300 mg C m$^{-3}$. This range differs from the zooplankton values at JdF shown in Figure 10c; this difference can be noted in the figure legend.

- L507*: "(F(2, 13) = 4.59, p < 0.05)"*.
  Please provide more details about this metric, either here when it is first mentioned or in the Materials and Methods section.

- L528: *"while the results presented here show similar patterns between model and observation data for certain parameters, these results varied depending on the depth ranges considered."*
  The analysis underlying this statement should be clarified, with an explicit reference to the supporting figure or table.

- Figures 12 and 13: Calculating this p-value over the periods pairwise (i.e., pre-NEP-MHW vs. NEP-MHW, and NEP-MHW vs. post-NEP-MHW) would have allowed quantification of the shifts between regimes. It is possible that in cases where the p-values over the entire period are not significant, a pairwise-period-specific p-value corresponding to a regime shift could have been significant.

**Minor comments**

- L47: The acronym SST (sea surface temperature) is defined later in the manuscript (L114). Please define it at its first occurrence (L47).
- L86: Please correct "JDF" to "JdF."
- L100: In the legend of Figure 1, a punctuation mark is missing between "boxes" and "Bathymetry." Please revise accordingly (i.e., "blue boxes. Bathymetry").
- L109: Punctuation "Suchy et al. ,2022 »
- You may use the acronym NEP-MHV that you previously introduced in the figure legends instead of spelling out "the Northeast Pacific marine heatwave" (e.g., L120: Figure 2; L256: Figure 3; L336: Figure 6), for consistency with the terminology used in the main text.
- L414: Fig. 9b, not 9c.

---

## Author Comment (AC1)

**General Comments**

We thank Anonymous Referee #1 for the careful and critical evaluation of our manuscript. We have reread our manuscript in the context of these comments and have thoroughly considered each suggestion which we believe will strengthen this work considerably. One of the repeating themes throughout Anonymous Referee #1's comments is that we have not been clear with our discussion of the source of nitrate anomalies as they relate to our two main regions of study (Juan de Fuca Strait [JdF] and Central Strait of Georgia [SoG]). This omission will be addressed by more clearly laying out the different dynamics specific to each of these regions in the Introduction, followed by more clarification in the Discussion. Another major concern raised by Anonymous Referee #1 is the inclusion of the NEP-MHW years in the climatology to which we compare our time series in our anomaly calculations. We address this concern as well as the line-by-line comments below.

G1. The paper mentions other contributing effects from El Nino Southern Oscillation (ENSO) and North Pacific Gyre Oscillation (NPGO). These events are known to affect regional weather/climate which in turn cause interannual variability in the meteorological and hydrological forcing. As a result, these events likely affect not only circulation and mixing, but also nutrient loads from rivers and streams, and upwelling. My concern is that there is no discussion of how changes in nutrient loads over pre-MHW, MHW, and post-MHW contributed to the biogeochemical (BGC) response, how loading changes were affected by MHW vs natural interannual variability.

Thank you for this comment. We agree that there is a lack of discussion about the effects of nutrient loads vs. natural interannual variability. While we already discuss the effects of El Niño events on upwelling and the subsequent impact on nutrient loading into the Salish Sea (L360-374), we will be strengthening this section as outlined below in our response to S4. P17., 365.

The revised manuscript will include a discussion on nutrient loading from rivers and streams in relation to the NEP-MHW years and natural interannual variability. Specifically, we will discuss the general patterns of nutrient loading via rivers and streams as well as the impacts of estuarine flow on nutrient enhancement. A previous modelling study of the Salish Sea by Khangaonkar et al., (2021) found that increased freshwater flow during the NEP-MHW years resulted in increased nutrients due to higher exchange flows relative to the pre-MHW reference year. Here our focus is the SoG; in the SoG, estuarine flow is a function of river flow (primarily the Fraser River), offshore upwelling, and the characteristics of the water currently sitting in the surface and intermediate layers in the SoG (Allen et al., 2025). In the present study we did not see high Fraser River discharge during the NEP-MHW years relative to the other years in our time series (Figure 2).

G2. The effect of MHW on BGC response has not been successfully teased out relative to natural variability that could be attributed to ENSO and NGPO. This is hard to do from analysis of observed data alone but is feasible using model sensitivity tests. The paper would be significantly strengthened with such an effort.

As Anonymous Referee #1 suggests, one of the major advantages of modelling studies is that sensitivity tests may be used to tease apart complicated processes and the subsequent biogeochemical responses. We used model experiments in our previous work examining the mechanistic link between the NPGO and plankton in the Central SoG (Suchy et al., 2025). However, after running a series of model experiments in that study it became evident that none of these warming signals acts alone, which is also one of the main takeaways from the current work. In order to effectively tease out the effects of the MHW relative to natural variability, such an analysis could look like running a number of years (ideally the entire time series) holding the boundary conditions at climatology conditions while letting the atmospheric conditions vary and then completing another model run wherein the boundary conditions are allowed to vary, but the same atmospheric conditions are held year after year. While we think this is a great idea, this type of analysis is beyond the scope of this study and would require considerable time (months of additional model runs) and computational costs. We believe that such an undertaking would warrant a whole paper on its own.

G3. In its present state the material presented does not allow the reader to clearly distinguish between cause and effect with respect to nutrient concentrations and plankton biomass in the upper 50 m. The climatological mean over the 16 years relative to which anomalies are discussed, includes NEP-MHW and other warming events.    I recommend a reanalysis relative to a new baseline with NEP MHW years excluded.

We understand Anonymous Referee #1's concern regarding calculation of the baseline which includes NEP-MHW years. We reanalyzed our data by calculating a climatology with the NEP-MHW years (2014-2017) removed. The additional figures shown below compare our anomaly results for the original climatology (all years) versus a climatology with the NEP-MHW years removed. The results show that our findings are robust even after removing the NEP-MHW years from the climatology. Given that our time series is 16 years long (2007 to 2022), our results show little sensitivity to how the climatology is defined.

**Temperature (left = original climatology; right = MHW years removed)**

[Figure]

**Nitrate (left = original climatology; right = MHW years removed)**

[Figure]

**Plankton (left = original climatology; right = MHW years removed)**

[Figure]

**Specific Comments**

S1. P3., 89: The flushing time of 47 days for a waterbody the size of Puget Sound seems too low. Is this value corroborated by other published literature?

> Here we are talking about flushing time as opposed to residence time. The value is from MacCready et al (2021) who note that, for Puget Sound, flushing time is about one third of the residence time and we will clarify this difference in the text. MacCready et al. 2021's value of 47 days from Table 1 is compared directly with another study (Sutherland et al. 2011) which gives a value of 57 so this value is similar to other studies previously reported.

S2. P13., 310: Negative Nitrate anomaly during NEP-MHW showing reduction in nitrate in surface layers is interesting and consistent. However, it is not clear if this is tied to stronger stratification and reduced mixing with surface layers or higher primary productivity from increased surface layer temperatures.

> The negative nitrate anomalies during the NEP-MHW were present through all depth layers, which was particularly evident in the JdF, Haro Strait, and Puget Sound regions. These results suggest a link to larger-scale processes such as upwelling near the open ocean boundary (related to El Niño events) for the Juan de Fuca region and NPGO-modulated nutrient variability which has been shown to be important in the Central SoG (Suchy et al., 2025). Looking at the JdF region, specifically, the strong nitrate anomalies were not associated with an increase in stratification. Phytoplankton biomass was slightly elevated but was not the highest observed throughout the time series. In our new version, a better framing of our two regions in the Introduction, as well as more careful discussion of the links between the climate indices and nutrients, will make the processes more clear. Specifically, we will highlight how the JdF region, with its direct connection to the open ocean is more strongly influenced by the shorter term (1-3 years) influences of SOI effects on the upwelling/downwelling of nutrients into the region. In contrast, the deep basin of the Central SoG is semi-enclosed and affected by regional scale forcing (from all of

the atmosphere, open ocean and rivers) that is integrated over longer time scales (such as the decadal time scales associated with the NPGO).

S3. P15. Figure 6 is about Nitrate anomalies, but caption text refers to temperature

Thank you. The figure caption will be corrected.

S4. P17., 365: Assessment of wind here would have helped strengthen this argument as it pertains to upwelling of nutrient rich waters into the estuary. Is reduced Northeasterly wind strength due to NEP-MHW or ENSO?

Our original statement here is confusing because the finding cited here (Hayashida et al., 2020) does not differentiate marine heatwaves from El Niño events. In general, a reduction in northeasterly wind strength is associated with ENSO. In the revised manuscript we will rewrite this statement to read: "Globally, marine heatwaves, in the form of El Niño events, have been shown to suppress nutrients in some regions by weakening the upwelling of nutrient-rich waters (Hayashida et al., 2020)." Winds directly relevant to upwelling on the west coast of Vancouver Island are outside of our SalishSeaCast model domain. However, we agree that this argument can be strengthened and will include a brief discussion of more recent studies related to upwelling/winds in the waters near the mouth of Juan de Fuca.

Papers to cite:

Fain & Peña 2025 (https://agupubs.onlinelibrary.wiley.com/doi/10.1029/2025JC022696)

Beutel et al., 2025 (https://doi.org/10.5194/bg-22-7309-2025)

S5. P18, 413: It is not clear from the data and modeling results whether Nitrate anomalies in Figure 10 are due to reduction in nutrients fluxes to the surface waters or because of increased phytoplankton consumption. As such we cannot speculate based on observed nutrient levels since effects of MHW are not isolated from other sources and sinks of nutrients. I suspect that simpler explanation is that nutrient concentrations in the surface layers are lower during MHW is because they have been consumed by higher primary productivity during that period. But then this argument fails during the post MHW period. Would it be possible to provide some clarification for this inconsistency between Pre- and Post-MHW

We agree with Anonymous Referee #1 that the lower nutrient levels in the observed 0-10 m nitrate appear to coincide with higher chlorophyll *a* concentrations, suggesting that the phytoplankton may be consuming more nutrients in this depth layer. The model outputs productivity and we see a maximum of 0.07 mmol N/m3/day increase in productivity during the NEP-MHW years compared to the climatology. Given a transit time of ~2.3 days (based on Thomson et al, 2007), we will show that extra nitrate draw down would be <15% which is insufficient to explain the nitrate decreases we see. This calculation will be included in the new version of the paper.

Indeed, as stated above, the model results show that the negative nitrate anomalies during the NEP-MHW occur throughout the water column and are linked to larger-scale processes. The JdF region is almost constantly mixed relative to the other regions. We will create new regional subplots to show this more effectively instead of including stratification results for all regions on one plot.

S6. P20., 432: What is the cause of this nitrate limitation in the upper 50 m. Is it reduced mixing (due to change in hydrodynamics and stratification) from the heatwave or is it a limitation caused by increased phytoplankton growth and consumption earlier in the year from higher temperatures.

*"Overall, nitrate was the most limiting to diatom growth during the summer months throughout the study, with persistent nitrate limitation occurring in the upper 50 m during all seasons from 2017 to 2022"*

Thank you for this comment as we now realize how poorly worded this statement was. A more accurate statement here would be: "Overall, nitrate was the most limiting to diatom growth during the summer months in the Central SoG. Nitrate limitation occurred periodically in the upper 50 m from 2017 to 2022, likely due to a combination of factors including increased stratification and weaker winds which prevented nutrients from being replenished into the surface waters (Moore-Maley & Allen 2022, Suchy et al., 2025).

S7. P25, Fig 12: Figure 12 is a good summary demonstrating that the model reproduces observed behavior, Pre-MHW, MHW, and Post-MHW years. Post MHW Nutrient levels in Juan De Fuca go up and are qualitatively supported by reduced growth relative to MHW, but they are still higher than pre-MHW. This indicates that other influences such as ENSO or NGPO which may be causing interannual variability may be at play. Is it possible for you to use the model to extract MHW effect from other influences.

It would be wonderful to use the model to extract individual influences of signals but to do so would require us to be able to extract the individual influences of signals in our forcing files: that is in the boundary conditions, atmosphere, and rivers. We have no idea how to do that and indeed, as, other than ENSO, these signals are empirical, not processes, it may not be possible to separate them even conceptually. Our findings show that the NEP-MHW co-occurs with the 2015-2016 El Nino, which contributes to part of the complication of the story. Our conclusions are that the NEP-MHW is associated with these other signals, not separate from them, which would make it almost impossible to pull the MHW out of the other forcing factors to run the model.

S8. P25, Fig.12: The chlorophyll and zooplankton during pre-MHW years are significantly lower than post-MHW while nutrient concentrations are not (Figure 12). This difference is not strongly reflected in model results (Figure 10), Is this simply due to lack of sufficient data or is there another process at play? It will be great if you could include a discussion on this noticeable difference. Also is it possible that pre-MHW results are influenced by the change

in source of Ocean Boundary Conditions after 2013 that was used in the model setup described previusly?

> We suspect that the low chlorophyll *a* and zooplankton during pre-MHW years in the JdF region is largely due to insufficient/patchy data (particularly for the zooplankton) and agree that a discussion on this noticeable difference should be added. Included in this discussion will be details about where in the water column the pre-MHW samples were collected as this information might have skewed the observation results low.

> Anonymous Referee #1 also raises a valid point about the change in boundary conditions. We have given considerable thought to the unfortunate timing of the change in the model's boundary conditions, i.e., in January 2013 which is just prior to our main time period of interest (2014-2017). However, we do not believe that the pre-MHW results were influenced by this change. First, we do not see evidence of a noticeable impact of the switch to LiveOcean boundary conditions in the temperature or nitrate results (Figures 3 and 6). Second, even if slight changes to the boundary conditions exist, we do not expect them to be large enough to propagate up the food web to the phytoplankton and zooplankton. Nevertheless, the revised manuscript will include this explanation in the context of our results.

S9. P27, L559: Could the authors provide justification for this statement. I may be misreading this but this statement seems to infer that decrease in nitrate during MHW is  caused by physical processes  independent of phytoplankton growth. If Salish Sea as the authors indicate is nutrient limited (due to healthy primary productivity), it is still nutrient limited with increased phytoplankton during MHW and lower nutrients could be a consequence.

> Original sentence:  *"Given that the waters in the JdF region are, on average, cooler and more nutrient-replete compared to other regions, the increased temperatures associated with the NEPMHW favoured phytoplankton growth and a resulted in a subsequent increase in zooplankton biomass, while the decrease in nitrate had little to no effect on the phytoplankton."*

.

> Thank you for this comment. Our original statement is poorly worded. In fact, the JdF region is almost always nutrient replete. Here we were attempting to explain that even during periods of negative nitrate anomalies, phytoplankton are not nitrate-limited in this region as nitrate concentrations remain, on average, >5 µM. "Given that the waters in the JdF region are nutrient replete and, on average, cooler compared to other regions, the increased temperatures associated with the NEP-MHW favoured phytoplankton growth and resulted in a subsequent increase in zooplankton biomass. In contrast, nitrate had little to no effect on phytoplankton growth as nitrate was always replete."

---

## Author Comment (AC2)

We thank Anonymous Referee #2 for the careful and critical evaluation of our manuscript. We have reread our manuscript in the context of these comments and have thoroughly considered each suggestion. The most substantial revision suggested by Anonymous Referee #2 would be the addition of a dedicated Discussion section to highlight the mechanisms behind the patterns we observed. We believe that these suggestions will greatly improve our manuscript. Specific comments are addressed below.

**Summary**

This manuscript describes the impacts of marine heatwaves on different regions of the Salish Sea. It examines the relationships between the Northeast Pacific marine heatwave (NEP-MHV; 2014–2017) and larger-scale climate signals such as the SOI and the NPGO. The study investigates the effects of temperature anomalies during these heatwave events on regional ecosystems, highlighting cascading effects across different phytoplankton and zooplankton groups. The results show that both the drivers of these perturbations and their ecological impacts vary among regions. The study combines modeling and observations, a comprehensive and relevant approach that supports the validity of the SalishSeaCast model, which performs well when compared with observations.
Overall, this work contributes meaningfully to our understanding of ecosystem perturbations associated with extreme climate events, an issue of growing importance as marine heatwaves are becoming more frequent and intense under ongoing global warming.
My main concern with the manuscript relates to the interpretation of the results. The processes underlying the contrasted responses observed across regions are complex but crucial to understand and to highlight. In its current form, these processes are not sufficiently interpreted or discussed, and the manuscript would benefit from a more thorough discussion of the mechanisms behind the observed patterns. The manuscript is well structured, clearly written, and highly relevant to the scope of the journal, provided that the following comments are addressed.

**General comments**

The manuscript does not include a dedicated Discussion section; elements of discussion are integrated in the Results section. However, interpretation of the impacts of the different marine heatwaves on biogeochemistry and lower trophic levels should be addressed in a dedicated section. It is essential to discuss the processes linking large-scale climate signals and local MHWs, as well as the relationships between temperature and nitrate anomalies and phytoplankton and zooplankton responses, including differences among the functional groups or the model.
The Discussion should address the following specific points:
•        Why are temperature anomalies in different regions more strongly correlated with certain large-scale climate signals than others (SOI, NPGO). The manuscript would benefit from presenting hypotheses, supported by relevant literature, to interpret these correlations.

Agreed (also see related comments made by Reviewer #1 above). We will do a better job of framing the different dynamics that influence the regions of interest in

the Introduction. Setting the stage for these regional differences will allow for a better discussion of our interpretation of the correlations with each region to the large-scale climate indices in a dedicated Discussion section.

For both sites (JdF and SoG), the manuscript should explain the cascading processes by which temperature and nitrate anomalies propagate through the food web, from phytoplankton to zooplankton, and discuss why/how these pathways differ between sites.

Thank you for this suggestion. We will improve our discussion of the cascading processes throughout the food web and how they differ between regions. If necessary, we will subtitle these sections to highlight these key differences. To summarize here, in the JdF region we observed increased warming during NEP-MHW. It is a "cooler" region relative to the SoG. During the NEP-MHW nitrate concentrations were lower than average but still not limiting (model and observations showed that concentrations were never <5 μM). Both diatoms and nanoflagellates in the model showed an increase in biomass due to the slight impact of temperature dependence on growth. Z1 responded well to this increase due to its feeding preference for both nanoflagellates and diatoms. In contrast, the Central SoG is one of the warmest regions of the Salish Sea. Warming was not as strong here during the NEP-MHW compared to the JdF region and was periodically interrupted by cooler waters during this time period. Nitrate is the most limiting factor to phytoplankton growth in this region with persistent nitrate limitation in summer affecting diatom growth due strong stratification and a lack of replenishment to the surface waters as a result of weaker winds throughout negative NPGO years. Nanoflagellates increased during the NEP-MHW. Z1 in this region followed closely the diatom pattern but the pattern for Z2 was not so clear.

Interpretation and discussion of the zooplankton results presented in Section 3.3.2 (see related comment below).

Detailed response provided below.

**Specific comments**
**Section 2.2 Study Period**
•        Throughout the manuscript, you refer to extended time periods associated with well-known large-scale climate signals, such as the NEP-MHV (2014–2017), SOI and El Niño events (e.g., 2010, 2015), and NPGO years (post-2014). However, these periods are described using different expressions, which may be confusing for the reader (e.g., L246: "post-2014 period (i.e., during negative or warm-phase NPGO years)"; L382, L453: "negative NPGO years"; L565: "NPGO negative (warm) years").

To improve clarity and consistency, it would be helpful to explicitly describe these climate events when they are first introduced in section 2.2, and to define a clear terminology that is then used throughout the manuscript. This approach is already well implemented for the NEP-MHV period (L103), and a similar treatment for the other climate signals would greatly

enhance readability. For example: "Monthly values for the NPGO index indicated a shift to the negative (warm) phase in October 2013 (Fig. 2). Hereafter, we refer to the post-2014 period as the 'negative NPGO years'."

> Thank you for this comment. We agree that clarification in the terminology here will improve readability. As such, we will make the suggested changes in the revised manuscript.

•          In several sections, key statements are supported almost exclusively by self-citations and are not sufficiently placed within the broader literature. For instance, in Section 2.2 (L108–108), relationships between NPGO/SOI and physical and biological conditions in the Salish Sea are referenced primarily through the authors' previous work, and in Section 3.1 (L274), relationships between temperature and large-scale climate indices in the Strait of Georgia are supported in a similar way. While these references are clearly relevant, including additional independent studies would strengthen the manuscript by better situating the results within the broader scientific context.

> Fair point. We will provide additional citations where relevant and will better situate some of our key statements within the broader scientific context.

**Section 2.3 SalishSeaCast Model**

•          L152: "Functional light dependence was switched to a potential energy curve but tuned to match the old response closely".

What do you mean by "tuned"? Please elaborate.

> We will revise the text to clarify what we mean by "tuned". The sentence will be revised to read: "Functional light dependence was switched to a potential energy curve with constants chosen to match the old response closely".

•          L174-187: Could you briefly describe the trophic interactions among the different model components (diatoms, nanophytoplankton, Z1, Z2)? This information would be useful for interpreting the cascading effects observed during marine heatwaves.

> Thank you for this suggestion. We are happy to add more detail on the trophic interactions among the different model components in the Methods section of the revised manuscript.

**Section 2.4 Model Data**
•          L195: "We focus our discussion mainly on the surface (0-50m) as this depth layer is most relevant to both phytoplankton and zooplankton.", and L202: "Model output for phytoplankton (diatoms and nanoflagellates) and zooplankton (Z1 and Z2) biomass were depth-averaged over the 0-50 m depth range to capture the full extent of the euphotic zone across regions (mmol N m-3)".

The manuscript would benefit from including information on the vertical structure and migratory behavior of zooplankton in the different regions of the Salish Sea. First, this would support the assumption that a 0–50 m layer is appropriate for representing zooplankton. In addition, such information would be valuable for interpreting the results on the impacts of marine heatwaves on lower trophic levels (cf. my comment on Section 3.3.2).

> We use 0-50 m because we wanted to balance where we know the phytoplankton are growing (euphotic zone) and where we know the model Z1 and Z2 are concentrated. For our two regions of focus, Juan de Fuca Strait and Central SoG, zooplankton are concentrated mainly in the upper 50 m (see Suchy et al. 2023). Furthermore, we will clarify that the surface layer is commonly defined by the 0-50 m depth layer in this region to clearly delineate between this layer and the underlying intermediate waters (e.g., Pawlowicz et al., 2007, Ianson et al., 2016). When model and observation data are considered together, we used full water column averages for zooplankton to avoid any confounding influence of the vertical structure and migratory behaviour of zooplankton on our results.

•        L204: "the extent to which temperature dependence and light/nutrient limitation was limiting to growth was calculated based on the phototrophic growth rate equations in the model".

Could you please elaborate on the methodology of this diagnostic?

> Absolutely. We agree with Anonymous Referee #2 that there is a lack of detail regarding the calculations of temperature dependence and light/nutrient limitation, which was evident in comments from both reviewers. The revised manuscript will provide an elaboration of this diagnostic as well as a more careful discussion of these results.

**Section 3.1.1 Temperature**
•        L271: "indicating a longer-term warming signal than can be explained by the NEP-*M_H_W_ _a_l_o_n_e_". _ _C_a_n_ _o_r_ _c_a_n_'t_ _?_ _*

> We will clarify this sentence to read: "...indicating a longer-term warming signal that cannot be explained solely by the NEP-MHW."

•        L275: "In contrast, surface temperature anomalies in the JdF region were significantly correlated to the SOI (r = -0.44, p < 0.001), but not to the NPGO (r = -0.11, p = 0.12; Table 1)." Could you provide an interpretation of this result, or propose hypotheses to explain it? See the general comment regarding the need for a dedicated Discussion section.

> Thank you. This comment is similar to a comment made by Anonymous Referee #1. We will provide an interpretation of this result in the revised manuscript. Our hypothesis is that the JdF region, with its direct connection to the open ocean is more strongly influenced by the shorter term (1-3 years) influences of SOI effects on the upwelling/downwelling of nutrients into the region. In comparison, the deep basin of the Central SoG is semi-enclosed and affected by regional scale forcing (from all

of the atmosphere, open ocean, and rivers) that is integrated over longer time scales such as the decadal time scales associated with the NPGO.

**Section 3.3.1. Phytoplankton**
• L411: "slightly less light limitation".

Please consider providing a metric or quantitative indicator to better support the interpretation of Figure 9. Moreover, why was this analysis/diagnostic performed only for diatoms? Applying a similar diagnostic to all four model components (diatoms, Z1, Z2) would help to interpret the processes underlying the variation patterns observed across the different trophic levels studied.

We focussed on diatoms with this metric because they are the most variable prey item for Z1 and Z2 in the model. A similar metric will be added for nanoflagellates to the supplemental material as light, temperature, and nutrients are less limiting to their growth in the model compared to diatoms; yet, they are still an important component of the Z1 and Z2 diet. To address Anonymous Referee #2's concern about the interpretation of this metric, we will include a better explanation of the figures. For example, Figure 9a is directly comparable to Figure 9b but this is not obvious to the reader (we separated the plots to make them less busy). A direct comparison of the two plots shows that temperature and light limitation are much larger in the JdF compared to either nutrient. In addition, we will add subplots to the right-hand side of Figure 9 showing the climatology for each limiting parameter (not the anomaly) to help the reader understand the magnitudes of the baselines to which anomalies are being calculated.

• L426: "Also evident was that diatoms during the post-2014 period (i.e., during negative or warm-phase NPGO years) peaked earlier, and for shorter durations, before switching to nanoflagellate dominance compared to the 2007 to 2014 period (Fig. 8b, Supp. Fig. S6a). »

Unclear, please clarify or rephrase. In particular, what is meant by "peaked earlier", earlier relative to what?

Here we mean that diatoms peaked earlier in the season relative to the pre-2014 period. This will be clarified in the text.

**Section 3.3.2. Zooplankton**
• In the JdF region, the marine heatwave has been described as being linked to the NEP-MHV and SOI, with the end of positive temperature anomalies and negative nitrate anomalies around 2020. From 2020 to 2022, negative temperature anomalies reappear, along with positive nitrate anomalies. A similar shift is observed for both nanophytoplankton and diatoms, with predominantly negative anomalies over the 2020–2022 period. Consequently, negative anomalies are also observed for Z1, but we do not observe the same response of Z2 (except for a peak in 2022). How can these contrasting responses between Z1 and Z2 be explained?

> Z1 biomass in the model follows closely with the diatom and nanoflagellate patterns because this class freely evolves based on model dynamics. Z2 is more constrained by the model's settings as it represents the model closure term. We will add this important clarification to Section 3.3.2 to avoid confusion.

More generally, providing information and relevant literature on the trophic structure and/or taxonomic composition in the different regions studied would greatly help to interpret these results. In particular, differences between the SoG and JdF sites, as well as communities' changes across the different periods within each site, should be discussed. Overall, this part of the results would benefit from being accompanied by interpretation in the Discussion section.

> Thank you for this comment. We will add information regarding the trophic structure and/or taxonomic composition where possible (i.e., in the Central SoG). Similar information for the JdF is more limited given the historical lack of zooplankton sampling in the region (see Figure 10c).

**Section 3.3.4. Model and Observation Comparisons**
•          L467: "model zooplankton (Z1 and Z2 combined) values were always within the range of observed values".

Figure 10c shows that the model does not capture the maximum observed zooplankton values. Please clarify.

> This statement will be revised to read: "That said, model zooplankton (Z1 and Z2 combined) values were always within the range of observed values although the model failed to capture some of the extreme values which we expect in the observation data due to the patchiness of zooplankton."

•          L470: "Both model nitrate and chlorophyll a showed better agreement with the observed values in the Central SoG".

Computing a comparison metric, such as the RMSE, for all variables in the observation–model comparison would help support this statement. This is not obvious for chlorophyll-a

> We agree that this statement is not obvious for chlorophyll *a*. In this paper we do not include point-by-point comparisons, thus making calculations of RMSE and other statistics less meaningful. Instead, we compared our model and observations over broader spatial scales. SalishSeaCast model has previously been evaluated extensively using point-by-point comparisons (depth and time matched) for nutrients (Olson et al., 2020) and chlorophyll *a* (Olson et al., 2020, Jarníková et al., 2022). While this sentence requires rewording, we will also add a section summarizing previous model evaluations in the Methods, which will include region-specific statistics (RMSE, bias, etc.).

- L486: Please reconsider the scale of Figure 11c, using a range of approximately 0–300 mg C m$^{-3}$. This range differs from the zooplankton values at JdF shown in Figure 10c; this difference can be noted in the figure legend.

  *Fair point and easy fix. The scale in Figure 11c will be adjusted and noted in the figure legend.*

- L507: "(F(2, 13) = 4.59, p < 0.05)".

Please provide more details about this metric, either here when it is first mentioned or in the Materials and Methods section.

  *Important comment. We will clarify in the methods that this metric presents results for the one-way ANOVA comparison of the mean values for pre-MHW, MHW, and post-MHW periods.*

- L528: "while the results presented here show similar patterns between model and observation data for certain parameters, these results varied depending on the depth ranges considered."

The analysis underlying this statement should be clarified, with an explicit reference to the supporting figure or table.

  *Thank you for this comment. We now realize this statement is somewhat misleading and also confusing. Choosing different depth ranges (e.g., 0-25 m vs. the 0-10 m for nitrate presented in this study) naturally affected the magnitude of the results presented. For example, nitrate concentrations averaged over the 0-10 m layer are lower than those averaged over 0-25 m. While the majority of the patterns we observed held true regardless of the depth layer chosen, there were some instances where significant differences between pre-MHW, MHW, and post-MHW were found when we used a different depth range even if significant differences were not found for the depth range presented in our results, and vice versa. Ultimately, the depths presented here were chosen so that we could maximize the number of observation samples available for comparison with the model. Nevertheless, this statement will be clarified in the revised manuscript.*

- Figures 12 and 13: Calculating this p-value over the periods pairwise (i.e., pre-NEP-MHW vs. NEP-MHW, and NEP-MHW vs. post-NEP-MHW) would have allowed quantification of the shifts between regimes. It is possible that in cases where the p-values over the entire period are not significant, a pairwise-period-specific p-value corresponding to a regime shift could have been significant.

  *The statistical analyses in Figs. 12 and 13 are one-way ANOVAs which test whether the means of the three groups are statistically different. The alternate hypothesis of this analysis is that at least one of the means is statistically different, but does not tell us which one(s) of means differ. Similar to what Anonymous Referee #2 is suggesting, a post-hoc pairwise corrected t-test can be used to tell us which set of means is different. Running these post-hoc t-tests confirmed the results from the*

one-way ANOVA. i.e., when there was a significant difference between the three groups, the pairwise test confirmed which periods (pre-MHW vs. MHW or MHW vs. post-MHW) was visually different on Figs 12 and 13. We will include the post-hoc paired t-test results in our revised manuscript.

**Minor comments**

•        L47: The acronym SST (sea surface temperature) is defined later in the manuscript (L114). Please define it at its first occurrence (L47).

Thank you. Sea surface temperature (SST) will be defined on L47.

•        L86: Please correct "JDF" to "JdF."

Thank you. Will be corrected.

•        L100: In the legend of Figure 1, a punctuation mark is missing between "boxes" and "Bathymetry." Please revise accordingly (i.e., "blue boxes. Bathymetry").

Thank you. This will be corrected.

•        L109: Punctuation "Suchy et al. ,2022 »

Thank you. Will be corrected.

•        You may use the acronym NEP-MHV that you previously introduced in the figure legends instead of spelling out "the Northeast Pacific marine heatwave" (e.g., L120: Figure 2; L256: Figure 3; L336: Figure 6), for consistency with the terminology used in the main text.

Thank you. Will be corrected.

•        L414: Fig. 9b, not 9c.

Thank you. This will be corrected.